# The V-ATPase *a*3 Subunit: Structure, Function and Therapeutic Potential of an Essential Biomolecule in Osteoclastic Bone Resorption

**DOI:** 10.3390/ijms22136934

**Published:** 2021-06-28

**Authors:** Anh Chu, Ralph A. Zirngibl, Morris F. Manolson

**Affiliations:** Faculty of Dentistry, University of Toronto, Toronto, ON M5G 1G6, Canada; anhnt.chu@mail.utoronto.ca (A.C.); ralph.zirngibl@utoronto.ca (R.A.Z.)

**Keywords:** V-ATPase, osteoclasts, bone, osteoporosis, osteopetrosis, anti-resorptive therapeutics, signalosome, TCIRG1, V-type proton ATPase 116 kDa subunit a3, OC-116 kDa, ATP6V0A3, ATP6V1C

## Abstract

This review focuses on one of the 16 proteins composing the V-ATPase complex responsible for resorbing bone: the *a*3 subunit. The rationale for focusing on this biomolecule is that mutations in this one protein account for over 50% of osteopetrosis cases, highlighting its critical role in bone physiology. Despite its essential role in bone remodeling and its involvement in bone diseases, little is known about the way in which this subunit is targeted and regulated within osteoclasts. To this end, this review is broadened to include the three other mammalian paralogues (*a*1, *a*2 and *a*4) and the two yeast orthologs (Vph1p and Stv1p). By examining the literature on all of the paralogues/orthologs of the V-ATPase *a* subunit, we hope to provide insight into the molecular mechanisms and future research directions specific to *a*3. This review starts with an overview on bone, highlighting the role of V-ATPases in osteoclastic bone resorption. We then cover V-ATPases in other location/functions, highlighting the roles which the four mammalian *a* subunit paralogues might play in differential targeting and/or regulation. We review the ways in which the energy of ATP hydrolysis is converted into proton translocation, and go in depth into the diverse role of the *a* subunit, not only in proton translocation but also in lipid binding, cell signaling and human diseases. Finally, the therapeutic implication of targeting *a*3 specifically for bone diseases and cancer is discussed, with concluding remarks on future directions.

## 1. Bone

Bone is a remarkable dynamic tissue which is involved in a variety of roles besides providing structural support. Bone exhibits endocrine, immune, mineral storage, growth factor, organ protection and repair functions [1,2,3]. Most of these functions can be attributed to the presence of three distinct major cell types, the osteoblast (OB), the osteoclast (OC) and the osteocytes. Osteoblasts are derived from mesenchymal stem cells during embryogenesis, and are responsible for the secretion of a proteinaceous matrix, including growth factors, which becomes mineralized [1,4]. OBs are found lining the bone surface and also become encased in the mineralized matrix, where they differentiate into osteocytes [5,6]. Osteocytes communicate with each other and other cell types via canaliculi found in bone. Osteocytes are capable of detecting stresses on the skeleton, and are able to activate OBs lining the bone surface, as well as OCs to start the repair process [7]. Osteoclasts have been thought to arise from hematopoietic cells exclusively; however, recent lineage tracing studies using mice have shown that there is also an extraembryonic component to this [8,9]. Cells derived from erythromyeloid-progenitors (EMP) in the embryonic yolk sac are the first wave of OC to differentiate, followed later by a distinct second wave derived from hematopoietic stem cells (HSCs). These two stem cell populations occupy two different niches in the adult, with the EMP homing to the spleen while the HSCs seed the bone marrow [8,9]. OCs are capable of resorbing bone via their ability to secrete acid to dissolve the mineral component and proteinases in order to digest the now exposed proteinaceous matrix [10,11]. This is a highly organized process that involves pre-OC cells fusing with each other, the formation of a sealed bone compartment underneath the now multinucleated OC sequestered by the sealing zone, and the formation of a ruffled membrane contained within the sealing zone [12,13,14]. The ruffled border acts as the gateway for the secretion of the acid and proteinases, and allows for the uptake of the dissolved mineral and digested proteins, which are mostly transcytosed by vesicles to the apical cell membrane for eventual disposal via the circulation [15]. Experiments performed in RAW 264 cells showed that the formation of an actin ring redirects intracellular vesicles, mostly secretory lysosomes, to transport large quantities of proteinases (e.g., Cathepsin K, alkaline phosphatase) and the acid generating machinery (made up from chloride channel 7 and the V-ATPase) to the OC plasma membrane adjacent to the bone surface [16,17]. Bone remodeling is a highly coordinated process that involves constant communication between OBs and OCs, and any interference with this can lead to disease [10,18,19]. V-ATPases are involved in pre-pro-protein processing (including glycosylation) [20], secretion [21], the internalization and degradation of molecules [22], vesicle transport and fusion [23,24], modulate signaling complexes, participate in distinct signalosomes [25], and promote cell migration in cancer [26]. To this end, mutations that interfere with V-ATPase function underlie diseases affecting a number of organ systems.

## 2. V-ATPase Functions

V-ATPases are ATP-driven proton pumps found in the endomembrane of the intracellular compartments in all of the eukaryotic cells and the plasma membrane of several specialized cells [27]. V-ATPases are responsible for acidifying and maintaining the pH of intracellular organelles, including the Golgi apparatus, endosome, lysosome and secretory vesicles [28,29]. V-ATPases pump protons into the Golgi apparatus, which become more acidic from the cis-Golgi to the trans-Golgi [20]. As newly synthesized proteins traverse the Golgi apparatus, they undergo post-translational modification including glycosylation, sulfation and phosphorylation. The maintenance of the pH gradient in the Golgi apparatus by V-ATPases is crucial for the function and localization of the glycosyltransferases required for the modification processes [30]. V-ATPase activity in the intracellular membrane is important for membrane trafficking processes such as receptor-mediated endocytosis [21,22]. The V-ATPase-dependent acidification of the endocytic compartments is required for the dissociation of ligand–receptor complexes, allowing the receptors to recycle to the cell surface. The released ligands are subsequently targeted to the lysosomes, where the low pH maintained by V-ATPases facilitates their degradation [31,32]. This process is important for the continued uptake of ligands such as low-density lipoprotein (LDL), a main carrier of plasma membrane cholesterol [33]. Many pathogens employ the V-ATPase-mediated acidification of the endocytic compartments to gain entry into cells, including diphtheria and anthrax toxins, as well as viruses such as influenza and Ebola [34,35]. After entering the host cells, viruses also require a low pH to trigger fusion and to deliver their viral genome into the host. V-ATPases are also involved in the intracellular trafficking of lysosomal enzymes by establishing a luminal pH gradient between compartments [21,32]. Lysosomes are more acidic than late endosomes, which in turn are more acidic than the trans-Golgi network (TGN). This gradient allows the binding of lysosomal proteases to the mannose-6-phosphate receptor at the TGN, facilitating the enzyme delivery to the lysosomes, and the dissociation of enzyme-receptor complexes in late endosomes, allowing the receptors to recycle to the TGN [36]. V-ATPases play a key role in cellular nutrient homeostasis by providing the acidic environment within lysosomes which is necessary for proteolysis, which is a major way in which cells generate free amino acids [37,38,39]. In addition to maintaining the lysosomal pH, V-ATPases also associate with the nutrient-sensing machinery in the lysosomal membrane, and are involved in the recruitment of the metabolic regulators mTORC1 and AMPK [38,39,40]. Within secretory vesicles, V-ATPases generate a proton gradient driving the uptake of small molecules such as the neurotransmitter glutamate [41,42,43], and they facilitate the processing of prohormones like proinsulin [33,44,45].

V-ATPases are targeted to the plasma membrane of specialized cells such as kidney intercalated cells [46,47], epididymis clear cells [48,49] and osteoclasts [50,51], where they function to transport protons from the cytoplasm to the extracellular space [30,52,53,54]. In the kidneys, V-ATPases are localized to the apical membrane of the alpha-intercalated cells to facilitate the secretion of protons into the urine in order to maintain pH homeostasis [54]. Osteoclasts rely on V-ATPases at the ruffle border for the demineralization of bone and the activation of the proteolytic enzymes required for bone resorption [55,56]. V-ATPases targeting the plasma membrane of epididymis clear cells are involved in the establishment of the acidic luminal pH necessary for sperm maturation and storage [49]. Recently, plasma membrane V-ATPases have been shown to be overexpressed in breast cancer cells, and to facilitate invasion by promoting the activity of acid-dependent proteases that degrade the extracellular matrix [57]. The inhibition of V-ATPases by concanamycin in prostate cancer cells results in a decreased level of mRNA for prostate-specific antigens [58]. Increasing evidence implicates the important role of V-ATPases in cancer cells’ growth and metastasis, and suggests a potential therapeutic treatment of metastatic cancer by the inhibition of V-ATPases activity.

In addition to the conventional functions of V-ATPases in intracellular signalling and membrane trafficking by generating pH gradients, recent findings suggest novel emerging roles of V-ATPases in the modulation of the function of receptors and their regulatory complexes through direct protein–protein interactions. For example, it was recently uncovered that Wnt/β-catenin signal transmission requires the interaction of co-receptor LRP6 with V-ATPase lysosomal accessory protein-2 (ATP6AP2) in late endosomes [59]. In Drosophila, V-ATPases have been suggested to be involved in the membrane fusion of synaptic vesicles via direct interaction with calmodulin [60]. Emerging studies propose the importance of V-ATPases in modulating various signalling pathways, including Notch, mTOR and AMPK via unconventional mechanisms [39,61].

In summary, V-ATPase-dependent acidification is essential for cellular metabolism, membrane trafficking and intracellular signalling. Moreover, the importance and novel emerging roles of V-ATPases in many signalling pathways and diseases, including cancers, makes them promising targets for drug development.

## 3. V-ATPase Structure

V-ATPases share their structure with mitochondrial and chloroplast F-type ATPases [27]. Both enzymes are composed of a peripheral catalytic sector (V_1_ or F_1_) and a membrane-bound proton channel sector (V_O_ or F_O_). They are evolutionarily related, and are functionally conserved as rotary proton pumps [44]. The eukaryotic V-ATPase is a 900 kDa complex consisting of sixteen subunits: A_3_B_3_CDE_3_FG_3_H comprising the V_1_ sector, and *ac_9_c*″ *def*AP1AP2 forming the membrane-bound V_O_ (the subscript numbers represent the subunits’ stoichiometry in the complex) (Figure 1) [62,63]. Subunits A and B are arranged in a hexameric configuration and contain the nucleotide binding sites responsible for ATP hydrolysis [64]. ATP hydrolysis creates a driving force to induce the rotation of the central stalk composed of subunits D, F and *d*, and the membrane-bound proteolipid *c*-ring *c_9_c*″ [65].

Each proteolipid subunit *c*, *c*″ has a conserved glutamate residue which is essential for proton translocation (E139 in *c*, and E98 in *c*″) [64,66,67]. The glutamate residues are protonated when the subunit rotates past the membrane-embedded C-terminal domain of the *a* subunit (*a*_CT_). The *a*_CT_ forms two half-channels that create a pathway for protons to cross from cytoplasm to the organellar lumen or the extracellular space [64]. Protons access the glutamate residue of subunit *c* upon entering the cytosolic half-channel, and the protonated glutamate residue carries the proton through the lipid bilayer as the *c*-ring rotates. The proton is released through the luminal half-channel following the deprotonation of the glutamate residue and stabilization by the critical arginine residue R740 in the *a* subunit (R735 in Vph1p, an S. cerevisiae ortholog of the *a* subunit) [68]. The AB hexamer is held stationary relative to the *a* subunit by three peripheral stalk EG heterodimers which connect the V_1_ sector to subunits C and H, and the N-terminal domain of the *a* subunit (*a*_NT_) [69].

V-ATPase activity is tightly controlled both spatially and temporally. One example of the temporal modes of V-ATPase regulation is the reversible assembly/disassembly upon environmental cues, which was first described in yeast [70]. The dissociation of V_1_-V_O_ sectors is regulated by nutrient availability, as the dissociated complex is inactive in both ATP hydrolysis and proton translocation, reflecting the cells’ attempt to conserve cellular ATP. In yeast, the dissociation occurs in response to glucose starvation, involves an intact microtubular network, and is reversible without new protein synthesis [71]. Upon V_1_–V_O_ disassembly, the C subunit dissociates from the V-ATPase complex, and the H subunit undergoes a conformational change resulting in the loss of the interaction with *a*_NT_ [72]. The reassembly of the complex requires the RAVE complex (Rav1, Rav2 and Skp1). The RAVE complex binds to subunits E and G, the dissociated C subunit of V_1_, and to the V_O_ subunit *a*, thereby positioning them to promote assembly [70]. The glucose-induced reassembly of V-ATPases requires the interaction of the protein complexes with regulatory proteins, such as the RAVE complex. Moreover, studies in yeast and RAW 264 osteoclast-like cells suggest a direct interaction between the glycolytic enzyme aldolase and V-ATPase subunits in a glucose-dependent manner [73]. The deletion of the aldolase gene in yeast resulted in V-ATPase disassembly and a reduction in V-ATPase activity [74]. In the presence of glucose, aldolase and V-ATPase interactions increase, inducing the reassembly of V_1_ and V_O_; hence, aldolase can act as a glucose sensor mediating V-ATPase assembly [75]. Several other determinants of V-ATPase assembly have been identified, including the membrane environment [76] and the interaction with regulatory factors such as HRG-1 [77] and viral infection [34]. The spatial regulation of V-ATPases is observed in the luminal pH gradients between compartments [78,79]. This mechanism of controlling V-ATPase activity is through the regulation of the trafficking of the complex, which is facilitated by different isoforms of the *a* subunit [80,81].

## 4. The V-ATPase *a* Subunit

Each V-ATPase complex contains one copy of the ~100 kDa *a* subunit, which exists as two isoforms in yeast (Vph1p and Stv1p) and four isoforms (*a1, a2, a3* and *a*4) in mammals [27,80,83]. The *a* subunit has a bipartite structure, with a cytoplasmic N-terminal half (*a*_NT_) and a membrane-integrated C-terminal half (*a*_CT_) which consists of eight transmembrane helices (Figure 2) [62,84]. As described above, two of the eight helices in *a*_CT_ are tilted and interact with the proteolipid *c*-ring to form the two hemichannels for proton translocation [62,85]. Even though ATP hydrolysis-coupled proton translocation can tolerate numerous *a* subunit mutations, the arginine residue in *a*_CT_ (R735 in Vph1p, and R740 in TCIRG1 encoding the mammalian *a*3 isoform) is absolutely essential [68]. The dominant R740S missense mutation of this critical arginine in mice uncouples the proton pumping activity from ATP hydrolysis, resulting in mice with a high bone mineral density [86]. The *a*_NT_*,* oriented parallel to the membrane, is essential for V-ATPase function as it couples V_1_ ATP hydrolysis to V_O_ proton translocation [44].

Studies with chimeric forms of Vph1p and Stv1p suggest that organelle targeting information is located in *a*_NT_ [87]. In yeast, V-ATPases are targeted to the vacuole and Golgi by Vph1p and Stv1p, respectively; when chimeric *a* subunits were made, the targeted organelles were determined by the *a*_NT_. Furthermore, mutagenesis studies revealed that the signal sequence W^83^KY within the *a*_NT_ of Stv1p is necessary for V-ATPase Golgi localization [88].

Similarly, in mammalian cells, different isoforms of the *a* subunit are enriched in specific organelles or cell types. However, the specific targeting signal of mammalian *a* isoforms has not been determined. V-ATPases containing the *a*1 isoform are found in the synaptic vesicles of neurons, and are relocated to the presynaptic plasma membrane at the nerve terminals [60,89]. The *a*2 isoform targets Golgi [90], and the *a*3 isoform is expressed in late endosomes and lysosome [16,91]. The *a*3 and *a*4 isoforms are also found on the plasma membrane of specialized cells, with *a*3 targeting the ruffle border of osteoclasts [50,92]; *a*4 is found in the apical membrane of kidney alpha intercalated cells and epididymal cells [49,93]. The upregulation of both *a*3 and *a*4 have been linked to the invasiveness of metastatic breast cancer cells [26]. Recently, the *a*4 isoform was shown to localize to the membrane of the invapodia of mouse breast cancer cells, where it plays a crucial role in the invasion and migration of the cancer cells [94]. While it is ubiquitously expressed in different organelles and cell types, the expression of *a*3 is approximately 100-fold greater in osteoclasts than in other cell types [95]. V-ATPases containing *a*3 are enriched in the membrane of the ruffled border, where they actively pump acid to dissolve bone and provide an acidic environment to activate the secreted proteases required for bone resorption. Furthermore, mutations in the *a*3 isoform in mammals—for example, the R740S in mice, mentioned above [86]—are associated with V-ATPase-related autosomal recessive osteopetrosis [53,96,97]. To this end, it is clear that the *a*3 isoform plays a crucial role in bone resorption by osteoclasts; therefore, the *a*3 isoform is a potential drug target for osteoporosis treatment, in which the excessive bone loss associated with this disease could be controlled by inhibiting *a*3-containing V-ATPases [50,98,99].

## 5. *a*3-*d*2-B2

Consistent with their diverse roles in intracellular compartments and different cell types, mammalian V-ATPase subunits have a variety of isoforms. Seven subunits have two to four isoforms, and the combination of different subunit isoforms are organelle– and tissue–specific [46,49,100,101]. For examples, the combination of *a*3, *d*2 and B2 is specific for osteoclasts [92,102,103]. Both *a*3 and B2 are highly expressed in the plasma membrane of osteoclasts [103], and studies in RANKL-differentiated RAW 264.7 cells indicate that *a*3 has a higher affinity for B2 than B1, and the inhibition of the *a3–*B2 interaction by the KM9114 compound may inhibit the resorptive activity of osteoclasts [55]. The *d*2 subunit has been found in various tissues, but is most abundant in osteoclasts [104], and a GST pull-down assay suggests a high-affinity interaction between *a*3 and *d*2 [102]. The RNAi knockdown of *d*2 resulted in the impairment of extracellular acidification by osteoclasts [105].

## 6. Human Diseases Linked to V-ATPase Mutations

As many biological processes are dependent on proton gradients, it is not surprising that close to half of the V-ATPase subunits are associated with human diseases; they are summarized in Table 1. While many disease-causing mutations within V-ATPase subunits have been identified, the precise molecular mechanisms underlying the ways in which mutations cause defects is generally not known. This is predominantly due to the lack of analysis other than reports of the mutations provided. Some mutations can be reconciled as null mutations (i.e., frame shifts, splice donor/acceptor mutations); however, most of these have not been experimentally demonstrated and rely heavily on computer prediction programs. For this review, we will restrict the discussion to mutations in the *a* subunit, with some general conclusions reached which are specific to this subunit.

First, most *a* subunit mutations are recessive, with a single wildtype gene being sufficient for V-ATPase function. For example, carrier parents with heterozygous mutations have no obvious phenotype, indicating that 50% gene expression is sufficient; the caveat to this statement is that reports on human *a*3 mutations rarely, if ever, thoroughly examine the heterozygous parents [97,108,109,110].

With respect to protein expression, patients with only 5–10% wildtype *a*3 protein expression can progress past the infantile mortality normally associated with *a*3 mutations; however, these patients still present with osteopetrosis, as well as vision and hearing loss [97,111,112]. Nevertheless, while dominant negative *a*3 mutations have not been identified in humans, our lab identified one example in mice; an R740S point mutation resulted in osteopetrotic mice in heterozygotes and early lethality in homozygotes. R740 is critical for proton translocation but not for folding, stability, or assembly. The *a*3_R740S protein is assembled into a complex and translocated to the OC ruffled border, but this results in an uncoupled enzyme with limited ATP hydrolysis and no proton translocation activity. These inactive V-ATPases displace active complexes, leading to the dominant negative phenotype [86,113].

While the R740S mutation is an example of a point mutation resulting in an assembled but inactive complex, it is more often the case that point mutations result in unstable and rapidly degraded *a* subunits [30,114,115]. One caveat to this general conclusion is that mutant protein stability is general assessed in in vitro cell culture systems, and the results for single mutations vary depending on the cell type and the presence/absence of endogenous protein. For example, it was shown that *a*3 point mutations with low expression levels in *a*3-null osteoclasts had wild-type expression in HEK293T cells [116].

Finally, V-ATPase-related diseases highlight specific cell types or cellular functions that are exquisitely dependent on specific isoforms, of which the function can’t be compensated for by the other paralogues, even if they are present in that cell [117]. This cell and/or organelle-specific dependence may be related to expression levels. For example, in yeast, where Vph1p is expressed at about 50 times the level of Stv1p, the absence of Vph1p results in a vacuolar pH defect despite the presence of Stv1p, but can be complemented by the overexpression of Stv1p [83]; this indicates that while isoforms can enzymatically complement each other, their expression levels must be adjusted accordingly in order to correct a phenotype. Similarly, in osteoclasts, the *a*3 expression is increased 100-fold during osteoclastogenesis. While *a*1 and *a*2 are expressed in osteoclasts, their expression levels cannot compensate for the absence of the highly expressed *a*3, hence the osteopetrotic phenotype. In intercalated kidney cells, B1, C1 and *a*4 subunits are highly expressed, with mutations in these isoforms resulting in renal tubular acidosis. In contrast, *a*2 is ubiquitously expressed in all cells, but is retained within Golgi. The cutis laxa phenotype resulting from *a*2 mutations suggests that while *a*1 and *a*3 expression may be equal to or exceed that of *a*2, their inability to be retained within Golgi is the limiting factor.

## 7. Membrane Signaling Lipids as Regulators of V-ATPase Localization and/or Activity

There is increasing evidence for the involvement of membrane-signaling lipids in V-ATPase regulation [76]. In yeast, a direct interaction between Vph1p, Stv1p and different membrane phosphoinositides (PIPs) has been shown, and was hypothesized to affect V-ATPase localization. Vph1p *a*_NT_ has been shown to interact with vacuolar membrane phosphoinositide PI(3,5)P_2_ in vitro; in the absence of PI(3,5)P_2_, Vph1p fails to localize to vacuoles [118]. On the other hand, Golgi-specific PI(4)P interacts with Stv1p *a*_NT_ to recruit Stv1p-containing V-ATPases to the Golgi, and enhances the activity of Golgi V-ATPases [119]. The Cryo-EM structure of V_O_ with Vph1p and Stv1p suggests binding sites for glycerophospholipids in both complexes [120]. These structural and biochemical studies suggest important roles for membrane lipids in V-ATPase function and a potential target for the modulation of its activity. The questions of whether human isoforms interact with PIPs, and whether such interaction affects the differential subcellular membrane distributions of V-ATPases in mammalian cells have not yet been determined.

## 8. V-ATPase as a Signalsome

As discussed above, V-ATPases are localized to numerous subcellular compartments with activities regulated to match the specific needs of each destination; how this destination-specific regulation is achieved is currently unknown, but details are emerging. Most V-ATPase localization studies have focused on the *a* subunit. However, another way to control V-ATPase subcellular localization is via the phosphorylation of the subunits [121]. In kidney cells, activation of PKA resulted in the phosphorylation of the Ser175 of the A subunit, which altered both V-ATPase subcellular localization and activity; in contrast, the phosphorylation of Ser384 by AMPK reduced the V-ATPase activity [122]. The direct binding of PKA or AMPK to any of the V-ATPase subunits has not been demonstrated. However, it has been shown that activated pPI3K, pAKT and pERK associate with the E subunit during virus replication [123,124,125]. A quick search through the PhosphoSitePlus database (phosphosite.org), a curated site for phosphorylation and ubiquitylation sites identified by high throughput mass spec analysis, revealed that all of the V-ATPase subunits are modified. Thus, there are clear indications that V-ATPases can bind various kinases, and that their activity/location is controlled by phosphorylation events.

Cellular compartments can be specified depending on which small guanine nucleotide binding protein is present on the membrane. The Rab subfamily is the largest, comprised of at least 60 different proteins. Rab proteins act as molecular switches, being ‘on’ in their GTP-bound form and able to bind effector proteins, and ‘off’ in their GDP-bound state [126,127]. This activity is controlled by two families of regulators, the Guanine Exchange Factor (GEF) that exchanges GDP for GTP, and Guanine Activating Protein (GAP), which activates the low intrinsic catalytic activity to promote GTP hydrolysis. *a*3 can bind to several Rab proteins directly in their GDP state, such as Rab7 (endosome, lysosome) and Rab27a [91]. This interaction has been mapped to the distal domain (Figure 2) in *a*3_NT_. This allows the *a*3 subunit to be relocated from the late endosome to the lysosome, and eventually to the ruffled border. Whether the other *a* subunit isoforms show distinct binding preferences for different Rab proteins has not been determined. The *a*2 subunit has also been demonstrated to inhibit Cytohesin2, the GEF for the Arf family of small GTPases, in a pH dependent manner. This function has been mapped to the N-terminus of *a*2, and is conserved in all of the isoforms of the *a* subunit [128]. This would allow V-ATPases to regulate subcellular trafficking by binding to small GTPases and potentially regulating GEF activity. Other GEFs which can be influenced by V-ATPases are Ragulator and Slc38A9, the GEFs for the Rag GTPases, which bring mTORC1 to the lysosome and activate it [129,130]. The mechanism of regulation may involve a direct interaction with *d*1 and the A/B subunit of active V-ATPases [39]. However, the detail mechanism is not yet understood. There are promising indications that mTOR may phosphorylate and regulate V-ATPase activity [37,39], but further investigation is required.

## 9. The Potential of *a*3 as a Therapeutic Target for Osteolytic Diseases

An advantage of targeting the V-ATPase *a*3 subunit to prevent osteolytic diseases is that osteoclast differentiation and fusion is unaffected by the absence of *a*3, as evidenced by the fact that, to date, all *a*3 mutations have resulted in osteoclast-rich osteopetrosis [96,131,132,133] (Figure 3, Table 2). The gold standard anti-resorptive treatments, bisphosphonates and anti-RANKL therapy, inhibit bone resorption by preventing osteoclast formation and/or triggering osteoclast apoptosis. While both are effective anti-resorptives, the resulting decrease in osteoclast numbers reduces osteoclast–osteoblast cell signaling, limiting the ability of combinational therapy with anabolic therapeutics to increase osteoblast bone formation [134,135]. We hypothesize that targeting *a*3 will inhibit the osteoclastic resorptive activity without affecting the osteoclast numbers, thus not interfering with osteoclast–osteoblast cross talk and subsequent osteoblastic bone formation. V-ATPase *a* subunits are not only attractive targets for osteolytic diseases, they are also considered therapeutic targets for the prevention of cancer, with increasing evidence that plasma membrane V-ATPases are required for extracellular acidification and subsequent metastasis [26]. V-ATPase are also therapeutic targets for fugal and viral infections; this is outside of the scope of this review, but was covered very well in a recent review [136]. V-ATPase-specific inhibitors such as bafilomycin and concanamycin induce apoptosis in all cell types. This highlights the fact that V-ATPase activity is essential for multiple cellular functions and the necessity to develop specific inhibitors targeting plasma membrane V-ATPases within osteoclasts and metastasizing cells. There are numerous studies looking at analogs of these plecomacrolides for derivatives with greater selectivity to osteoclast V-ATPases; this is, again, outside the scope of this review, but is well summarized in the recent review by Duan et al. [98]. Strategies for the targeting of *a*3-containing V-ATPase can be grouped into three general categories: preventing cell-specific protein–protein interactions, targeting extracellular domains, and gene therapy.

Several groups, including our own, are working on targeting protein–protein interactions unique to V-ATPases on the plasma membrane of the osteoclast. Using the yeast two-hybrid assay, our group identified a direct protein–protein interaction between the *a*3 and *d*2 subunits, both of which are isoforms which are highly expressed in osteoclasts. We recreated this interaction in vitro with heterologously expressed proteins and used high-throughput screening to look for compounds that would inhibit the *in vitro d2* interaction with *a*3, but not with the other *a*1, *a*2 and *a*4 orthologs [102]. We were then able to show that one of these compounds, luteolin, reduced the osteoclastic resorption without affecting the osteoclast viability or actin ring formation [102]. Holliday’s group found interactions between the V-ATPase V_1_ B2 subunit and actin, and demonstrated that this interaction was essential for transporting V-ATPases to the osteoclast plasma membrane, which is essential to resorption. They computationally modeled this interaction in silica and performed a virtual screen for inhibitory compounds, resulting in the identification of enoxacin [137,138]. Holliday and others have subsequently shown that enoxacin and its derivative, bis-enoxacin, can prevent bone resorption in animal models [139,140].

A second strategy is to generate inhibitory antibodies to epitopes located on the extracellular/luminal loops of subunit “*a*”. This approach takes advantage of the fact that these domains should only be exposed to the extracellular surface when osteoclasts are actively resorbing bone, or when cancer cells are metastasizing and V-ATPases are localized on the plasma membrane. This approach is facilitated by the fact that the extracellular domains have been clearly defined both through biochemistry [141] and through structural analysis using cryo–electron microscopy [62]. A recent report showed that monoclonal antibodies against the V-ATPase *a*2 subunit delayed ovarian tumor growth [142]. The inhibitory antibody was generated against amino acids 488–510 in human *a*2, which the authors stated is in the “transmembrane region of the protein” [143]. Nevertheless, the most recent structures of *a*1 predict that residues 488–510 of human *a*2 would be in the large extracellular/luminal loop between transmembrane domains 3 and 4 [62]. Further, our group has shown that *a*2 is glycosylated at residues N484 and N505 [82], providing biochemical evidence that this epitope is luminal/extracellular and thus accessible to anti-*a*2 antibodies in culture media, supporting the overall strategy described above.

Finally, the use of gene therapy to decrease *a*3 expression has been shown to be effective in decreasing both bone resorption and cell metastasis in cell culture and animal models. Hu et al. used small interfering RNA (siRNA) directed against *a*3 and found reduced bone resorption in a rat osteoclast culture [144]. Jiang et al. locally injected an *a*3-specific adeno-associated virus-mediated small-hairpin RNA (shRNA) into the periodontal tissues in vivo, and reported that it protected mice from *P. gingivalis* infection-stimulated bone resorption. In the same paper, they also reported that haploinsufficient Atp6i^+/−^ mice were similarly protected from *P. gingivalis* infection-stimulated bone loss [145].

With respect to the prevention of metastasis, siRNAs and small-hairpin RNA (shRNA) specific to *a*3 reduced the invasiveness of MCF10CA1a [146] and B16-F10 melanoma cells [147], respectively, while siRNA specific to both *a*3 and *a*4 inhibited the invasion of MB231 cells [148]. In contrast, Flinck et al. found that knocking down *a*3 increased the migration and transwell invasion of pancreatic ductal adenocarcinoma cells, leading the authors to conclude that *a*3 negatively regulates migration and invasion [149].

To summarize, the data from in vitro and in vivo models suggest that the reduction of *a*3 expression or *a*3-specific interactions can prevent bone loss and cell metastasis. While encouraging, clinical trials with *a*3 targeted therapeutics have yet to be reported.

## 10. The Potential of the Correction of *a*3 Splice Site and Missense Mutations to Treat Osteopetrosis

As mentioned above and summarized in Figure 3, mutations in *a*3 account for over 50% of infantile malignant autosomal recessive osteopetrosis. The majority of *a*3 mutations predict no or severely truncated versions of *a*3, leaving hematopoietic stem cell transplantation as the only current therapeutic option available. While effective, it must be performed during infancy for a successful outcome. Children diagnosed at an older age with milder forms of osteopetrosis currently have no treatment option, but the milder disease suggests the limited expression of functional *a*3. Our own study examined a 7 year old child with osteopetrosis resulting from a silent mutation in a conserved splice site motif [97]. The aberrant splicing reduced the full-length wildtype *a*3 expression to approximately 5–10%, explaining the phenotype but leaving the child without treatment options. Additional silent splice site mutations have been reported in older children with osteopetrosis [111]. While these children are not recommended for hematopoietic stem cell transplantation, the identification of aberrant splicing as the root cause suggests treatments focused on increasing the full length mRNA expression using specific small interfering RNAs [150] or splice-switching antisense oligonucleotides [151].

Similarly, the identification of specific missense mutations and understanding the precise molecular mechanisms underlying the mutation could help to inform a rational drug design. To this end, our group explored two human *a*3 mutations (G405R and R444L) by recreating them in the yeast ortholog, Vph1p (G424R and R462L) [115]. We found that both mutations did not affect the subunit expression, assembly or localization, but reduced hydrolytic rate and proton translocation, suggesting that these residues are critical to enzymatic activity and not amenable to rescue. In an alternate approach, our group used human embryonic kidney (HEK) transiently transfected with plasmids expressing *a*2 and *a*4 mutations identified in cases of cutis laxa and renal tubular acidosis, respectively. Two of the four mutations—*a*2 P405L and *a*4 R449H—affected protein stability and subsequent ER retention and degradation [30]. The identification of the folding/stability/ER exit issues as the primary molecular defect opens up the possibility of using chemical chaperones to stabilize the folding, thus escaping degradation and allowing ER exit.

## 11. Future Directions

Table 2 lists the splice site, missense and small deletion *a*3 mutations resulting in osteopetrosis. As indicated in the table, the majority of these mutations were identified by genomic sequencing with no subsequent analysis, as access to patient tissue is understandably limited. Nevertheless, our group demonstrated different in vitro [30,114,115] and in vivo approaches with limited patient tissue [97] to elucidate the molecular mechanism underlying the disease-causing mutations. Performing these analyses helps elucidate the *a*3 residues critical to folding, ER exit, assembly, targeting and activity, and as detailed above, can open up therapeutic possibilities for osteopetrotic patients. The further identification of critical residues, regions and interactions unique to *a*3 could also open up the possibility of screening for compounds to inhibit *a*3 towards therapeutics for osteolytic diseases and cancer metastasis. Critical residues in predicted luminal/extracellular domains open up the possibility of inhibitory antibodies that could only access plasma membrane V-ATPase complexes. As we hope this review has highlighted, the analysis of *a*1, *a*2 and *a*4 orthologs informs *a*3 structure and function. As mentioned above, any therapeutic targeting of *a*3 must be specific to plasma membrane *a*3. To this end, it is critical that we understand where the targeting information resides, but this essential information is still not known. With respect to *a*3 being a therapeutic target to prevent metastasis, similar to our approach targeting the osteoclast-specific *a3-d2* interaction [102], the identification of cancer-specific subunit interactions could be informative. To this end, a recent paper did indeed look for cancer specific “V-ATPase molecular signatures” in a variety of different tumor cells [152]. Finally, as noted above, haploinsufficient Atp6i^+/−^ mice were protected from bone loss in a bacterial infection-stimulated model of periodontal disease [145]. This result is at odds with the fact that both heterozygote mice and humans with only one wild type copy of *a*3 are asymptomatic, but could reflect that in a disease/stressed state, the gene copy number becomes critical. Given the results of Jiang et al. [145], it would be of interest to note whether heterozygote *a*3/- individuals are similarly protected against bone loss in inflammatory arthritis, periodontal disease and/or postmenopausal osteoporosis.

## Figures and Tables

**Figure 1 ijms-22-06934-f001:**
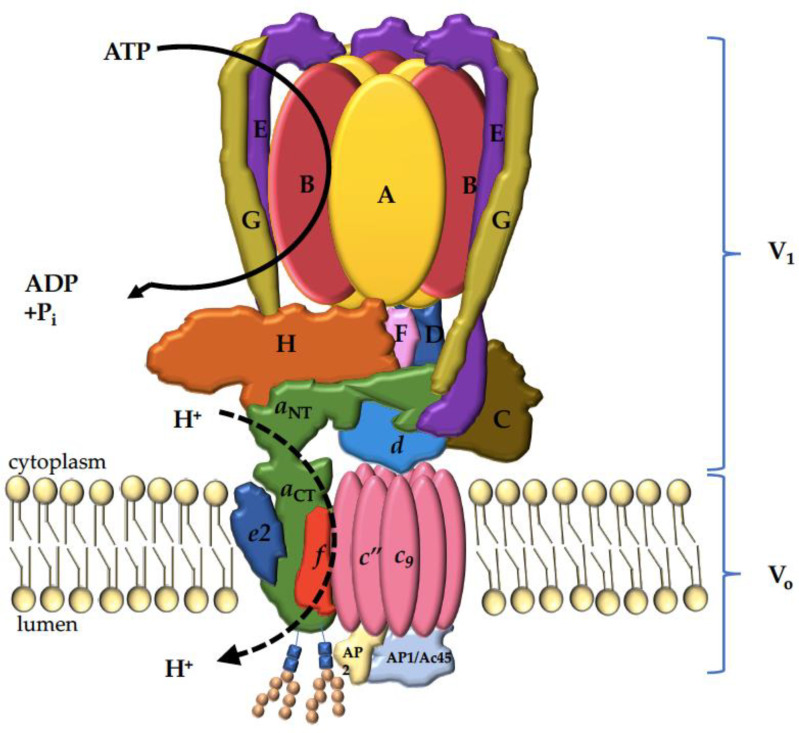
Mammalian V-ATPase complex. Cytosolic sector V_1_, comprised of A_3_B_3_CDE_3_FG_3_H, is responsible for the ATP hydrolysis, which generates the force required to drive the rotation of the proteolipid *c*-ring (*c*_9_*c*″) of the membrane-bound V_O_ consisting of *ac*_9_*c*″*def*AP1AP2. The *a*_CT_ forms two half-channels that create a pathway for protons to cross the lipid bilayer as the *c*-ring rotates. Both *a*2 and *a*3 orthologs are glycosylated twice on the first luminal loop within the C-terminus (depicted here), whereas *a*1 and *a*4 are only glycosylated once [82].

**Figure 2 ijms-22-06934-f002:**
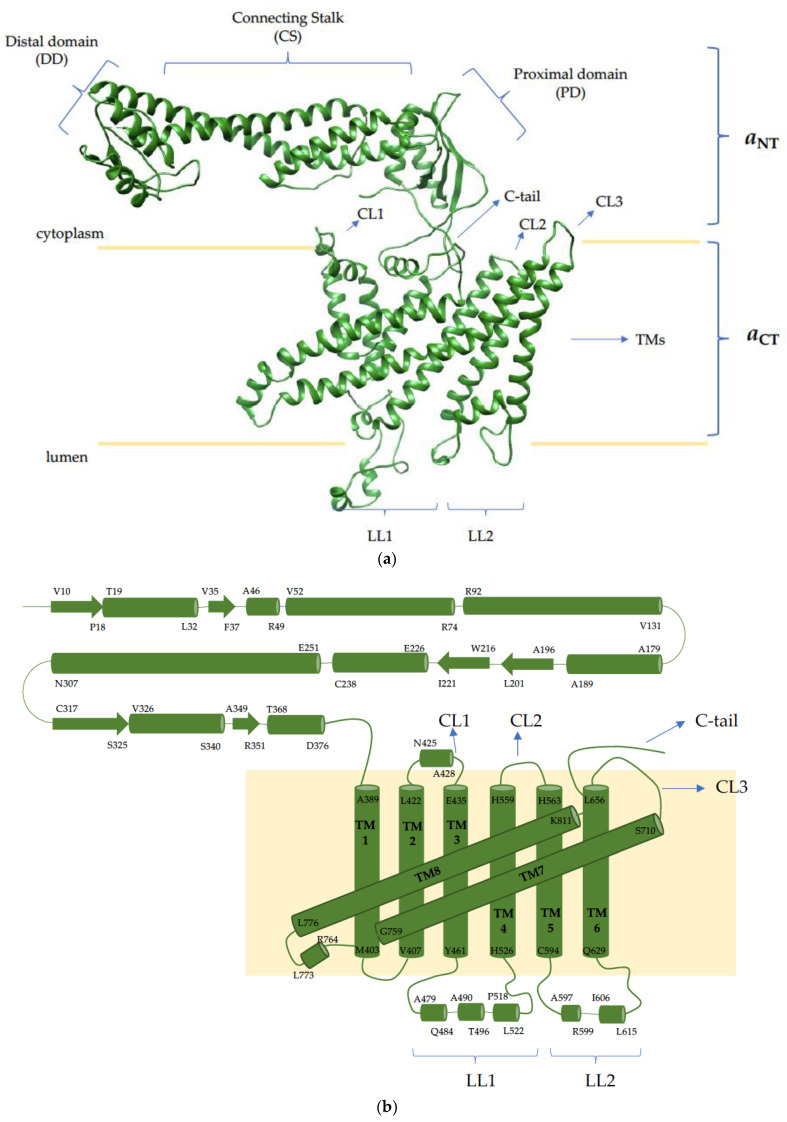
Mammalian V-ATPase *a*3 subunit. (**a**) Homology model of the *a*3 isoform generated using the Phyre2.0 server with constraint coordinates from the mammalian brain *a*1 isoform (PDB: 6vqc_3); (**b**) topology of the *a*3 isoform. The *a* subunit contains a cytoplasmic N-terminal half (*a*_NT_), which can be divided into three sub-domains—a distal domain (DD), connecting stalk (CS) and a proximal domain (PD)—and a membrane-bound C-terminal half (*a*_CT_) consisting of eight transmembrane helices (TM1-8), two of which are tilted and form the two hemichannels with the proteolipid *c*-ring. Cytosolic loops (CL1-3) connect TM2 and 3, TM4 and 5, and TM6 and 7, respectively; luminal loops 1 and 2 (LL1 and LL2) connect TM3 and 4, and TM5 and 6, respectively. Within luminal loop 1, *a*2 and *a*3 orthologs are glycosylated twice, whereas *a*1 and *a*4 are glycosylated once [82].

**Figure 3 ijms-22-06934-f003:**
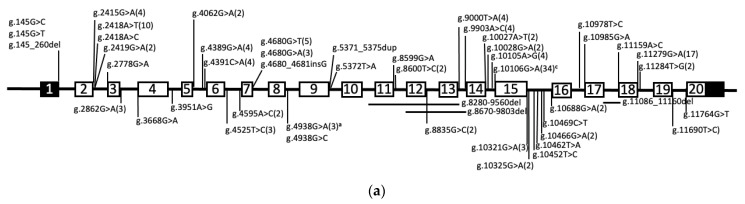
Mutations in *TCIRG1* cause malignant autosomal recessive osteopetrosis (Table 2). (**a**) Splice-site mutations in the introns of *TCIRG1.* (**b**) Missense and small deletion mutations in the *a*3 isoform. ^a^ Founder effect in the Chuvashiya and Marians population; ^b^ founder effect in the Costa Rica population; ^c^ founder effect in the Flanders population.

**Table 1 ijms-22-06934-t001:** Summary of V-ATPase subunit nomenclature, gene designation and associated human diseases.

Subunit Designation ^a^	Yeast Gene ^b^	Human Gene ^b^	Function	Human Disease (OMIM#) ^c^
**V_1_ Subunits**				
A	VMA1	ATP6V1A	ATP binding/hydrolysis	#617403-Cutis laxa, type IID (AR) #618012 Developmental and epileptic encephalopathy 93 (AD)
B	VMA2	ATP6V1B1	ATP binding/hydrolysis	# 267300-Renal tubular acidosis, distal, with progressive nerve deafness (AR)
		ATP6V1B2		# 124480-Deafness, congenital, with onychodystrophy; DDOD (AD) # 616455-Zimmermann-Laband syndrome 2; ZLS2 (AD)
C	VMA5	ATP6V1C1		
		ATP6V1C2		Recessive renal tubular acidosis [106]
		ATP6V1C3		
D	VMA6	ATP6V1D	Torque transmission	
E	VMA4	ATP6V1E1	Stator function	# 617402-Cutis laxa, type IIC; ARCL2C (AR)
		ATP6V1E2		
F	VMA7	ATP6V1F	Torque transmission	
G	VMA10	ATP6V1G1	Stator function	
		ATP6V1G2		
		ATP6V1G3		
H	VMA13	ATP6V1H		
**V_O_ Subunit**				
a	VPH1/STV1	ATP6V0A1	Stator function, proton transport	Developmental and epileptic encephalopathy [107]
		ATP6V0A2		# 219200-Cutis laxa, type IIA; ARCL2A(AR) # 278250-Wrinkly skin syndrome; WSS (AR)
		TCIRG1		# 259700-Osteopetrosis, type B1; OPTB1 (AR)
		ATP6V0A4		# 602722-Renal tubular acidosis, distal; RTADR (AR)
c	VMA3	ATP6V0C	Rotation, proton transport	
C′	VMA11			
c″	VMA16	ATP6V0B		
d	VMA6	ATP6V0D1	Transmission of torque, coupling ratio	
		ATP6V0D2		
e	VMA9	ATP6V0E1		
		ATP6V0E2		
f	YPR170W-B	RNASEK		
Ac45	VOA1	ATP6AP1	V_O_ assembly	# 300972-Immunodeficiency 47; IMD47 (XLR)
M8-9		ATP6AP2	V_O_ assembly	# 301045-Congenital disorder of glycosylation, type IIr; CDG2R (XLR) # 300423-Mental retardation, X-linked, syndromic, hedera type; MRXSH (XLR) # 300911-Parkinsonism with spasticity, X-linked; XPDS (XLR)
**Chaperone**				
VMA12	VPH2	TMEM199	Stabilizes Vph1p, binds Vma22p	# 616829-Congenital disorder of glycosylation, type IIp; CDG2P (AR)
	VMA21	VMA21	Assembles V_O_, exports V_O_ from ER to Golgi	# 310440-myopathy, X-linked, with excessive autophagy; MEAX (XLR)
	VMA22	CCDC115	Stabilizes Vph1p, binds Vma12p	# 616828-Congenital disorder of glycosylation, type IIo; CDG2O (AR)

^a^ Historical subunit designation; ^b^ official gene symbol http:/ncbi.nlm.nih.gov/gene (May 2021); ^c^ online Medelian inheritance in man (OMIM) http:/www.omim.org (May 2021).

**Table 2 ijms-22-06934-t002:** Summary of the described *TCIRG1* mutations in osteopetrotic patients.

Genomic ^a^		cDNA ^b^	Protein ^c^		Allele Freq. ^d^	Comment. [Ref]
Domain	Position	Position	Domain	Position		
Int1	g.145G>C				1	[153]
Int1	g.145G>T				1	[153]
Int1	g.145_260del				1	[153]
E2	g.2363del	c.66del	PD	p.Y23Tfs*4	2	Listed as c.65(E2):delC; p.22A>Afs5 [112,154]
Int2	g.2415G>A	c.117+1G>A			4	[155,156,157]
	g.2418A>T	c.117+4A>T	PD	p.V26_D39del	10	Activates cryptic splice site in E2 resulting in a deletion. [96,156,157,158,159]
Int2	g.2418A>C	c.117+4A>C			1	[108]
Int2	g.2419G>A	c.117+5G>A			2	[154,156]
Int2	g.2778G>A	c118-1G>a			1	[153]
Int3	g.2862G>A	c.196+5G>A			3	[154]
Int3	g.3668G>A	c.197-1G>A			1	[153]
E4	g.3714del	c.242del	CS	p.P81Rfs*85	1	[160]
Int4	g.3951A>G	c.418-21A>G			1	[154]
E5	g.3975G>C	c.421G>C	DD	p.A141P	1	[156]
E5			DD	p.G159Rfs*68	1	Listed with no details [161]
E5	g.4034dup	c.480dup	DD	p.P161Afs*66	2	Listed as c.475dupC [153]
Int5	g.4062G>A	c.503+5G>A			2	[162]
Int5	g.4389G>A	c.504-8G>A	DD	p.N168Kfs*55 and/or p.N168Kfs*8	4	Activates cryptic splice in E6 resulting in a 11bp deletion of E6 and/or skipping of E6 [156]
Int5	g.4391C>A	c.504-6C>A	DD	p.N168Kfs*55 and/or p.N168Kfs*8	4	Activates cryptic splice in E6 resulting in a 11bp deletion of E6 and/or skipping of E6 [53,158,163]
E6	g.4406_4407delinsTA	c.514_515delinsTA	DD	p.G172Y	1	[164]
E6	g.4517del	c.624del	DD	p.V209*	2	Described as p.P208Pfs*1 [165]
E6	g.4523G>A	c.630G>A	DD	p.T210	3	We showed exon skipping and activation of cryptic splice site [162]
Int6	g.4525T>C	c.630+2T>C			3	[156,157,158]
Int6	g.4595A>C	c.631-2A>C			2	[154]
E7	g.4613G>A	c.647G>A	DD	p.W216*	1	[155]
E7	g.4614_4640del	c.648_674del	DD	p.W216_G225delinsC	1	[153]
E7	g.4615_8590del	c.649_1297del	DD	p.M217Rfs*95	2	Deletes most of E7 into E11 [154]
E7	g.4622T>C	c.656T>C	DD	p.F219S	1	[166]
E7	g.4637G>A	c.671G>A	DD	p.G211_W224del	1	Activates splice acceptor in E7 resulting in deletion [53]
E7	g.4651G>T	c.685G>T	DD	p.G229*	1	[154]
E7	g.4654C>T	c.688C>T	DD	p.Q330*	2	[157]
E7	g.4658del	c.692del	DD	p.K231Rfs*48	1	[112]
E7	g.4668del	c.702del	DD	p.I235Sfs*44	6	[156,167]
E7	g.4679G>T	c.713G>T	DD	p.C238F	1	Assumes no alt. splicing as this is the last nt of E7 [53]
Int7	g.4680G>T	c.713+1G>T			5	[157,159]
Int7	g.4680G>A	c.713+1G>A			3	[156,168,169]
Int7	g.4680_4681insG	c.713+1_713+2insG	DD	p.C238Wfs*252	1	Assumes no alt splicing [153]
E8	g.4851A>G	c.725A>G	DD	p.H242R	1	De novo mutation, not other mutation found, potential dominant [170]
E8	g.4909del	c.783del	CS	p.Q261Hfs*18	1	[153]
E8	g.4922G>T	c.796G>T	CS	p.E266*	1	[154]
E8	g.4923del	c.797del	CS	p.E266Gfs*12	1	[171]
Int8	g.4938G>A	c.807+5G>A	CS	p.L271Gfs*231	23	Founder mutation in Chuvashiya population. Resulted in activation of cryptic splice donor 37nt downstream. [164,172]
Int8	g.4938G>C	c.807+5G>T			1	[153]
E9	g.5181_5186delinsA	c.831_836delinsA	CS	p.F277Lfs*211	1	[173]
E9	g.5233C>T	c.883C>T	CS	p.Q295*	1	[153]
E9	g.5259C>A	c.909C>A	CS	pY303*	2	[112,174]
E9	g.5272del	c.922del	CS	p.E308Sfs*4	2	[53,156]
E9	g.5321_5322insG	c.971_972insG	PD	p.C324Wfs*166	2	[157]
E9	g.5329C>T	c.978C>T	PD	pR327*	1	[153]
E9	g.5357del	c.1007del	PD	p.L336Rfs*10	2	[155]
E9	g.5357_5363del	c.1007_1013del	PD	p.L336Pfs*8	2	[154]
E9	g.5365A>T	c.1015A>T	PD	p.S339C	1	[154]
E9	g.5369_5370insGGTGA	c.1019_1020insGGTGA	PD	p.M341Vfs*7	1	Described as p.340S>Sfs151 [154]
Int9	g.5371_5375dup	c.1020+1_1020+5dup			1	[165]
Int9	g.5372T>A	c.1020+2T>A			1	[153]
E10	g.5988G>T	c.1024G>T	PD	p.E342*	1	[96]
E10	g.6000_6001dupGTGC	c.1037_1040dup	PD	p.V348Cfs*143	1	Described as c.1036_1037insGTGC [154]
E10	g.6078G>T	c.1114C>T	PD	p.Q372*	2	[154,160]
E10	g.6078C>G				1	listed as p.Q372* [157], but would be pQ372E if it is g.6078C>G
E10	g.6082del	c.1118del	PD	pG373Afs*30	1	[153]
	g.8280_9560del	c.1166_1554del	TM1	p.A389Dfs*151	4	Deletion includes E11-13 [156,158,168]
E11	g.8464_8465insA	c.1171_1174insA	TM1	p.Y391Ifs*99	2	Listed as g.8464insA [153,157]
E11	g.8484del	c.1191del	TM1	p.F398Sfs*5	2	Listed as c.1188delC; p.P397Pfs6 [154]
E11	g.8489T>G	c.1196T>G	TM1	p.L399R	1	[153]
E11	g.8506G>A	c.1213G>A	TM1	p.G405R	23	Founder mutation effect in Costa Rica [112,153,154,156,157,162,168]
E11	g.8521G>A	c.1228G>A	TM2	p.G410R	1	[168]
E11	g.8521G>T	c.1228G>T	TM2	p.G410W	1	Listed as p.G410R [154]
E11	g.8521G>C	c.1228G>C	TM2	p.G410R	1	[153]
E11	g.8523del	c.1230del	TM2	p.L411Cfs*19	5	Listed as Pt1 delG8521; p>G410fsX429 or Pt20 g.8521delG [158,168,169]
E11	g.8542G>A	c.1249G>A	TM2	p.A417T	1	[153]
E11	g.8548_8549insGG	c.1255_1256insGG	TM2	p.A419Gfs*12	1	[153]
E11	g.8569C>T	c.1276C>t	CL1	p.R426*	3	[153,175]
E11	g.8590C>T	c.1297C>T	CL2	p.Q433*	5	[156,157]
E11	g.8598G>C	c.1305G>T	TM3	p.E435D	1	[153]
Int11	g.8599G>A	c.1305+1G>A			1	[153]
Int11	g.8600T>C	c.1305+2T>C			2	[110,159]
	g.8670_9803del	c.1306_1554del	TM3	p.Q438_W520del	1	Deletion includes E12-13. Checked both DNA and cDNA [153]
E12	g.8695del	c.1328del	TM3	p.G443Afs*85	1	[153]
E12	g.8698G>T	c.1331G>T	TM3	p.R444L	6	Founder effect in Cost Rica population [157]
E12	g.8716T>G	c.1349T>G	TM3	p.M450R	1	[153]
E12	g.8738del	c.1371del	TM3	p.G458Afs*70	6	Listed as c.1370delc; p.T457Tfs71 [154]
E12	g.8738C>A	c.1371C>A	TM3	p.I436Afs*70	2	Creates cryptic splice acceptor, results in the deletion of 67nt of E12 [176]
E12	g.8739G>A	c.1372G>A	TM3	p.G458S	3	[153,154]
E12			TM3	p.F459Lfs*79	1	No other details provided [161]
E12	g.8749_8751del	c.1382_1384del	LL1	p.N462del	4	[156,157]
E12	g.8755delinsGCTTCATCTACAACG	c.1387delinsGCTTCATCTACAACG	LL1	p.E463Gfs*70	1	Listed as c.1387insGCTTCATCTACAACG; pGlu463Glyfs [171]
E12	g.8759C>A	c.1392C>A	LL1	p.C464*	3	[53,156,157]
E12	g.876-_8766del	c.1393_1399del	LL1	p.F465Afs*61	1	[153]
E12	g.8788C>A	c.1421C>A	LL1	p.S474*	5	[156,167]
E12	g.8795G>A	c.1428G>A	LL1	p.W476*	1	[153]
E12	g.8799_8816delinsT	c.1432_1449delinsT	LL1	p.V478Sfs*6	1	[153]
E12	g.8805_8806del	c.1438_1439del	LL1	p.A480Dfs*9	2	[96]
E12	g.8807del	c.1430del	LL1	p.M481Wfs*47	2	[177]
Int12	g.8835G>C	c.1463+5G>C			2	[178]
E13	g.8952_8553insA	c.1507_1508insA	LL1	p.N503Kfs*167	2	[179]
E13	g.8980C>A	c.1536C>A	LL1	p.Y512*	6	[156,158,169,180]
E113	g.8993G>A	c.1549G>A	LL1	p.D517N	3	[156,181]
Int13	g.9000T>A	c.1554+2T>A			4	[157,182]
Int13	g.9903A>C	c.1555-2A>C			4	[154,168]
E14	g.9909G>A	c.1559G>A	LL1	p.W520*	1	[110]
E14	g.10003_10004insGTGG	c.1653_1654insGTGG	TM4	p.L552Vfs*119	1	Listed as RT p.V451fsX670 [182]
Int14	g.10027A>T	c.1673+4A>T	TM4	p.V558Afs111	2	Not obvious how V558 becomes A [164]
Int14	g.10028G>A	c.1673+5G>A			2	Retains int14 or skips E14/15 [156,157]
Int14	g.10105A>G	c.1674-2A>G			3	Appears as if intronic seq are retained [162]
Int14	g.10106G>A	c.1674-1G>A			34	Founder mutation if Flanders population [53,156,157,159,168,169]
E15	g.10115delinsTT	c.1682delinsTT	LL2	p.G561Vfs*109	1	[168]
E15	g.10117C>T	c.1684C>T	LL2	p.Q562*	1	[182]
E15	g.10166T>G	c.1733T>G	TM5	p.L578R	1	[154]
E15	g.10168	c.1735G>A	TM5	pG579R	1	[153]
E15	g.10208G>A	c.1775G>A	TM5	p.W592*	1	[154]
E15	g.10220G>A	c.1787G>A	LL2	p.W596*	2	[96]
E15	g.10242_10251del	c.1809_1818del	LL2	p.P604Sfs*80	1	[163]
E15	g.10270_10273del	c.1837_1840del	LL2	p.M613Sfs*63	1	[154]
E15	g.10311C>A	c.1878C>A	LL2	p.Y625*	2	[168]
Int15	g.10321G>A	c.1887+1G>A			3	[154]
Int15	g.10325G>A	c.1887+5G>A			2	Listed as c.1941+5G>A. Makes 32% normal transcript [111]
Int15	g.10452T>C	c.1887+132T>C			1	[110]
Int15	g.10462T>A	c.1887+142T>A			1	retains 6.7% wt splicing [110]
Int15	g.10466G>A	c.1887+146G>A			2	retains 5.5% wt splicing [110]
Int15	g.10469C>T	c.1887+149C>T			1	[110]
Int15	g.10688G>A	c.1888-1G>A			2	Listed asc.1874-1G>A [168]
E16	g.10692del	c.1891del	TM6	p.V631Wfs*56	1	Listed as c.1878delG [153]
E16	g.10698C>T	c.1897C>T	TM6	p.E633*	1	[154]
E16	g.10735_10736insGGCA	c.1934_1935insGGCA	TM6	p.I646Afs*25	2	[166]
E16	g. 10809C>T	c.2008C>T	CL3	p.Arg670*	26	Founder variant in Flander population [154,156,157,168,174]
Int16	g.10978T>C	c.2014-8T>C				[157]
Int16	g.10985G>A	c.2014-1G>A				[153]
E17	g.11049G>T	c.2077G>T	CL3	p.E693*	1	[153]
E17	g.11086_11160del	c.2114_2119-1del	CL3	p.E706Rfs*123	2	Deletes last 5nt of E17 and all if Int17 [157]
Int17	g.11159A>C	c.2119-2A>C			1	[153]
E18	g.11172_11190del	c.2130_2148del	TM7	p.E711Pfs*24	1	[153]
E18	g.11195del	c.2153del	TM7	p.I718Tfs*23	1	[156]
E18	g.11202_11204del	c.2160_2162del	TM7	p.I721Rfs*109	5	[182]
E18	g.11203_11205del	c.2161_2163del	TM7	p.I721del	4	[154,156,166]
E18			TM7	pI721Afs*14	2	No data shown [161]
E18	g.11223C>A	c.2181C>A	TM7	p.C727*	3	[112,154]
E18	g.11227T>C	c.2185C>T	TM7	p.S729P	1	[153]
E18	g.11240C>T	c.2198C>T	TM7	p.S733F	1	[153]
E18	g.11260_11261del	c.2218_2219del	TM7	p.L740Qfs*90	4	[154,168]
E18	g.11278C>T	c.2236C>T	TM7	p.Q746*	5	Assumes splicing to E19 [159,169,181,183].
Int18	g.11279G>A	c.2236+1G>A			17	PCR analysis shows minor wt transcript, and at least 5 alternative splice events [156,157,158,162,168,169]
E19	g.11556del	c.2282del		p.G761Afs*22	2	[153]
E19	g.11558_11559insC	c.2284_2285insC		p.L762Pfs*69	1	[153]
E19	g.11598C>G	c.2324C>G	TM8	p.P775R	6	[156,157,168]
E19	g.11602_11610del	c.2328_2336del	TM8	p.F777_A779del	1	[157]
E19	g.11622T>C	c.2348T>C	TM8	pM783T	2	[110,158]
E19	g.11646T>G	c.2372T>G	TM8	p.M791R	1	[153]
E19	g.11647_11650del	c.2376_2379del	TM8	p.Q792Dfs*28	2	[157,158]
E19	g.11651G>C	c.2377G>C	TM8	p.G793R	1	[168]
E19	g.11654_11655del	c.2380_2381del	TM8	p.A796Lfs*34	1	[108]
E19	g.11657_11658del	c.2383_2384del	TM8	p.A796Lfs*34	1	[168]
Int19	g.11690T>C	c.2414+2T>C			1	[173]
Int19	g.11764G>T	c.2115-1G>T			1	[155]
E20	g.11765G>A	c.2415G>A	TM8	p.W805*	1	Assumes correct splicing. This is 1st nt in E20 [96].

^a^ Genomic reference sequence: AF033033.2; ^b^ cDNA reference sequence: NM_006019.3; the A in the ATG start codon is designated as 1. ^c^ Protein reference sequence: NP_006010.2. ^d^ Allele frequency: we define this as the number of times the allele has been described in the literature, with a homozygous allele being counted as two, and a heterozygous allele being counted as one. Dom-domain: for genomic sequences (E, Exon; Int, Intron) and for protein domains (PD, proximal domain; CS, connecting stalk; DD, distal domain; TM, transmembrane domain; CL, cytoplasmic loop; LL, luminal loop; CT, cytoplasmic tail). Note: The authors wish to stress the importance of reporting the correct version number when reporting mutations. For TCIRG1 in particular, reporting the reference sequence as MN_006019 is insufficient, as this is associated with two proteins. MN_006019.1 is linked to NP_006010.1, which is 829 amino acids long. MN_006019.2, MN_006019.3, MN_006019.4 and AF033033.2 (Genomic) are linked to NP_006010.2, which is 830 amino acids long due to the Ala603 in exon15. This leads to confusion, especially from multicentre studies, in which the reported mutations after Ser602 and c.1806 don’t agree within the same publication because different reference sequence versions were used by the labs. Our table has been adjusted to reflect this difference.

## Data Availability

Not applicable.

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
