# Peer review of "The V-ATPase a3 Subunit: Structure, Function and Therapeutic Potential of an Essential Biomolecule in Osteoclastic Bone Resorption"

_ijms, 2021, doi:10.3390/ijms22136934_

Round 1
Reviewer 1 Report
Chu et al reviewed the structure, function and potential medical applications of the V-ATPase a3 subunit. It is determined that this review can efficiently deliver information to researchers or readers related to V-ATPase a3. I endorse the publication of this review in the journal after making several corrections below.
1. Figures 1 and 2a: aNT and aCT are underlined in red. Author need to remove it.
2. Figure 2b: Author should align the positions of amino acids in the figure.
3. Please add references to the sentences below.
-line 33-37: “Osteoblasts are derived from mesenchymal stem cells during embryogenesis and are responsible for secreting a proteinaceous matrix, including growth factors, which becomes mineralized. OBs are found lining the bone surface and also become encased in the mineralized matrix where they differentiate into osteocytes.”
-line 42-48: "Cells derived from erythromyeloid-progenitors (EMP) in the embryonic yolk sac are the first wave of OC to differentiate followed later by a distinct second wave derived from hematopoietic stem cells (HSCs). These two stem cell populations occupy two different niches in the adult, with the EMP homing to the spleen while the HSCs seed the bone marrow. OCs are capable of resorbing bone via their ability to secrete acid to dissolve the mineral component and proteinases to digest the now exposed proteinaceous matrix."
-line 255-259” First, most a subunit mutations are recessive with a single wildtype gene sufficient for V-ATPase function. For example, carrier parents with heterozygous mutations have no obvious phenotype, indicating that 50% gene expression is sufficient; the caveat to this statement is that reports on human a3 mutations rarely if ever thoroughly examine the heterozygous parents.”
-line 278-281: “Finally, V-ATPase related diseases highlight specific cell types or cellular functions that are exquisitely dependent on specific isoforms, whose function can’t be compensated for by the other paralogues, even if they are present in that cell. This cell and/or organelle-specific dependance may be related to expression levels.”
-line 309-313: “We identifed a critical motif in the N-terminal of isoforms a2 and a4 which are required for interaction with PIPs in vitro and for membrane association of the proteins in HEK293 cells; moreover, mutations of the amino acids in the putative motif have been found in patients with cutis laxa and distal renal tubualar acidosis, respectively.”
-line315-319: “V-ATPases are localized to numerous subcellular compartments with activity regulated to match the specific needs of each destination; how this destination-specific regulation is achieved is currently unknown, but details are emerging. Most V-ATPase localization studies have focused on the a subunit. However, another way to control V-ATPase subcellular localization is via phosphorylation of the subunits.”
-line 325-329: “A quick search through the PhosphoSitePlus database (phosphosite.org), a curated site for phosphorylation and ubiquitylation sites identified by high throughput mass spec analysis, reveals that all of the V-ATPase subunits are modified.”
-line 330-334: “Cellular compartments can be specified depending on which small guanine nucleotide binding protein is present on the membrane. The Rab subfamily is the largest, comprised of at least 60 different proteins. Rab proteins act as molecular switches, being ‘on’ in their GTP bound form and able to bind effector proteins, while being ‘off’ in their GDP bound state.
-line 353-358: “An advantage of targeting the V-ATPase a3 subunit to prevent osteolytic diseases is that osteoclast differentiation and fusion is unaffected by the absence of a3 as evidenced by the fact that to date all a3 mutations result in osteoclast-rich osteopetrosis (Figure 3, Table 2). The gold standard anti-resorptive treatments, bisphosphonates and anti-RANKL therapy, inhibit bone resorption by preventing osteoclast formation and/or triggering osteoclast apoptosis.”
-line 378: 384: Several groups, including our own, are working on targeting protein-protein interactions unique to V-ATPases on the plasma membrane of osteoclast. Using the yeast two-hybrid assay, our group identified a direct protein-protein interaction between the a3 and d2 subunits, both isoforms highly expressed in osteoclasts. We recreated this interaction in vitro with heterologously expressed proteins and used high through-put screening to look for compounds that would inhibit the in vitro d2 interaction with a3, but not with the other a1, a2 and a4 orthologs.“
4. line 307-309: “Our lab is currently working on characterization of a putative PIPs binding domain within the aNT, and is investigating the role of this domain in membrane targeting/retention of the whole V-ATPase complex.”
-I do not think this sentence is appropriate for a review paper as it does not report research results. If the author wishes to add this sentence, the author must add a validated reference.
5. Figure 2 caption: “Mammalian a3 subunit.” Should be “Mammalian V-ATPase a3 subunit.”
Reviewer 2 Report
In the manuscript “The V-ATPase a3 subunit: structure, function, and therapeutic potential of an essential biomolecule in osteoclastic bone resorption” by Chu, et al, the role of the a3-subunit of V-ATPase, and more generally a-subunit orthologs and paralogs are examined. The a3- subunit is highly expressed in osteoclasts and crucial for bone resorption but the underlying mechanism by which a3, and other a-subunits, function is not fully understood. The article does an excellent job of establishing the importance of a3, surveying the literature to establish what is known, and areas in need of further study, and of looking at possible therapeutic possibilities that arise from the crucial and selective role the a3 subunit plays. I recommend publication after a few minor additions. 1. I think it is appropriate to cite “Blair HC, Teitelbaum SL, Ghiselli R, Gluck S. Osteoclastic bone resorption by a polarized vacuolar proton pump. Science. 1989 Aug 25;245(4920):855-7. doi: 10.1126/science.2528207. PMID: 2528207,” as the original identification of V-ATPase in the osteoclast ruffled membrane at some point. 2. I have concerns regarding reference 11. This, very influential article, article makes use of RAW 264.7 osteoclast-like cells, a cell type that does not (as far as I can tell) make ruffled membranes. The article shows an electron micrograph of a3-labeling in a “resorbing osteoclast” that does not display a ruffled membrane and the “pit” looks to me like ridges of knife marks on the bone slice. The point I think should be made that this was done is RAW cells and not primary osteoclasts and that this is a weakness. 3. In the introduction, “two distinct cell types”, is a bit confusing. Osteocytes are included as differentiated osteoblasts. I think however that the findings that osteocytes are by far the primary stimulators of osteoclastic resorption, under normal conditions, is an important point and should be clarified. Nakashima T, Hayashi M, Fukunaga T, Kurata K, Oh-Hora M, Feng JQ, Bonewald LF, Kodama T, Wutz A, Wagner EF, Penninger JM, Takayanagi H. Evidence for osteocyte regulation of bone homeostasis through RANKL expression. Nat Med. 2011 Sep 11;17(10):1231-4. doi: 10.1038/nm.2452. PMID: 21909105. Xiong J, Onal M, Jilka RL, Weinstein RS, Manolagas SC, O'Brien CA. Matrix-embedded cells control osteoclast formation. Nat Med. 2011 Sep 11;17(10):1235-41. doi: 10.1038/nm.2448. PMID: 21909103; PMCID: PMC3192296. 4. While the nutrient sensing articles cited are important, I think that the finding that aldolase interacts with V-ATPase in yeast and humans, including osteoclasts and is linked to glucose sensing is worth mentioning: see refs below. Lu M, Holliday LS, Zhang L, Dunn WA Jr, Gluck SL. Interaction between aldolase and vacuolar H+-ATPase: evidence for direct coupling of glycolysis to the ATP-hydrolyzing proton pump. J Biol Chem. 2001 Aug 10;276(32):30407-13. doi: 10.1074/jbc.M008768200. Epub 2001 Jun 8. PMID: 11399750. Lu M, Sautin YY, Holliday LS, Gluck SL. The glycolytic enzyme aldolase mediates assembly, expression, and activity of vacuolar H+-ATPase. J Biol Chem. 2004 Mar 5;279(10):8732-9. doi: 10.1074/jbc.M303871200. Epub 2003 Dec 12. PMID: 14672945. Zhang, CS., Hawley, S., Zong, Y. et al. Fructose-1,6-bisphosphate and aldolase mediate glucose sensing by AMPK. Nature 548, 112–116 (2017). https://doi.org/10.1038/nature23275 Li M, Zhang CS, Zong Y, Feng JW, Ma T, Hu M, Lin Z, Li X, Xie C, Wu Y, Jiang D, Li Y, Zhang C, Tian X, Wang W, Yang Y, Chen J, Cui J, Wu YQ, Chen X, Liu QF, Wu J, Lin SY, Ye Z, Liu Y, Piao HL, Yu L, Zhou Z, Xie XS, Hardie DG, Lin SC. Transient Receptor Potential V Channels Are Essential for Glucose Sensing by Aldolase and AMPK. Cell Metab. 2019 Sep 3;30(3):508-524.e12. doi: 10.1016/j.cmet.2019.05.018. Epub 2019 Jun 13. PMID: 31204282; PMCID: PMC6720459. 5. In discussing the rationale for targeting a3, or the osteoclast V-ATPase, for therapies treating bone disease, the following reference is relevant: Karsdal MA, Martin TJ, Bollerslev J, Christiansen C, Henriksen K. Are nonresorbing osteoclasts sources of bone anabolic activity? J Bone Miner Res. 2007 Apr;22(4):487-94. doi: 10.1359/jbmr.070109. PMID: 17227224. In general, this is a strong and very interesting article addressing potential new approaches to treating bone disease. It is well written and the figures illustrative. I recommend it for publication in IJMS after the suggestions I have made are considered.Author Response
Please see the attachment

Round 2
Reviewer 1 Report
The authors address all concerns in the manuscript. The currently revised manuscript is suitable for publication on IJMS. I support the manuscript for publication.